# Unraveling Survival Determinants in Patients with Advanced Non-Small-Cell Lung Cancer with *EGFR* Exon 20 Insertions

**DOI:** 10.3390/curroncol32030174

**Published:** 2025-03-18

**Authors:** Kung-Yang Wang, Shih-Chieh Chang, Yu-Feng Wei, Jui-Chi Hung, Chung-Yu Chen, Cheng-Yu Chang

**Affiliations:** 1Division of Chest Medicine, Department of Internal Medicine, Far Eastern Memorial Hospital, New Taipei City 220, Taiwan; b98401011@gmail.com (K.-Y.W.); ellyhung1206@gmail.com (J.-C.H.); 2Division of Chest Medicine, Department of Internal Medicine, National Yang Ming Chiao Tung University Hospital, Yi-Lan 260, Taiwan; 11319@hosp.nycu.edu.tw; 3School of Medicine for International Students, College of Medicine, I-Shou University, Kaohsiung 824, Taiwan; yufeng528@gmail.com; 4Department of Internal Medicine, E-Da Cancer Hospital, I-Shou University, Kaohsiung 824, Taiwan; 5Department of Internal Medicine, National Taiwan University Hospital Yunlin Branch, Yunlin County 640, Taiwan; 6Department of Electrical Engineering, Yuan Ze University, Taoyuan City 320, Taiwan

**Keywords:** non-small-cell lung cancer, rare *EGFR* Mutations, exon 20 insertions

## Abstract

Background: Lung cancer is the leading cause of cancer-related death in Taiwan. It is often associated with mutations in the epidermal growth factor receptor (*EGFR*) gene, with common mutations accounting for approximately 85% of all *EGFR*-related cases. However, the remaining 15% are caused by uncommon mutations in *EGFR*, mainly insertions in exon 20 (about 4%). The response to EGFR tyrosine kinase inhibitors (TKIs) can vary markedly with exon 20 insertions. However, few prior large-scale studies have examined patients with these *EGFR* mutations. Methods: This study combines the databases of several large hospitals in Taiwan to analyze the effects and clinical significance of rare EGFR mutations on responses to EGFR-TKIs, considering the changes in medication. Results: This study enrolled 38 patients with non-small-cell lung cancer and *EGFR* exon 20 insertions. It assessed the correlations of various predictors with progression-free survival (PFS) and overall survival (OS). It showed that among those with *EGFR* exon 20 insertions, the median PFS was 5.15 months, and OS reached 13 months. The median PFS was 5.4 months for afatinib, 5.7 months for chemotherapy, and 4.3 months for first-generation EGFR-TKIs. Conclusions: EGFR-TKIs may be considered as an alternative treatment option for patients with *EGFR* exon 20 insertions in cases where the currently recommended therapies, such as chemotherapy with or without amivantamab, are either unavailable or intolerable. The potential use of afatinib for specific patients in this context depends on the precise characteristics of their mutation and remains to be determined.

## 1. Introduction

According to data from the Ministry of Health and Welfare, lung cancer was the leading cause of cancer-related death in Taiwan in 2023. Molecular testing and mutation analysis have significantly transformed the treatment of advanced non-small-cell lung cancer (NSCLC), which is associated with high mortality. It is often also associated with mutations in the epidermal growth factor receptor (*EGFR*) gene, with common mutations accounting for approximately 85% of all *EGFR*-related cases [1], including exon 19 deletions and the L858R mutation in exon 21, which both respond well to *EGFR* tyrosine kinase inhibitors (TKIs) [2,3,4]. However, the remaining 15% of mutations in *EGFR* primarily consist of insertions in exon 20, point mutations in exon 18, and complex compound mutations. Responses to EGFR-TKIs can differ significantly due to the presence of uncommon mutations. These mutations can affect treatment efficacy and patient outcomes, highlighting the importance of genetic testing to tailor therapies for individual cases and improve overall effectiveness in cancer management [5,6,7].

Prior to this study, large-scale research on patients with uncommon *EGFR* mutations could be traced back to a national study conducted at Taipei Veterans General Hospital in 2015 [8]. However, since the approval of afatinib for use in patients with uncommon *EGFR* mutations by the US Food and Drug Administration in 2018, the landscape may have changed regarding survival outcomes and the clinical significance of this mutation for Taiwanese patients with NSCLC. In Taiwan, due to health insurance coverage, many patients with NSCLC and exon-20 insertions receive afatinib as a first-line treatment, and a significant number of patients also undergo first-line chemotherapy.

This study aimed to examine treatment responses in Taiwanese NSCLC patients with exon 20 insertions, the most common type of uncommon *EGFR* mutations. It combines data from the databases of several large hospitals in Taiwan to examine the effects and clinical significance of this *EGFR* mutation on responses to *EGFR*-TKIs in recent years, considering changes in medication.

## 2. Patients and Methods

(1)Study design: This multicenter, retrospective study was conducted at a top-tier medical center (a hospital providing advanced and specialized care) and three regional hospitals in Taiwan.(2)Patients: We included all patients with stage IV NSCLC who had EGFR exon 20 insertion mutations and received treatment at our institution between January 2016 and December 2020. However, patients who were referred only for best supportive care or those who participated in clinical trials were excluded. The mutational status of all included patients was determined using polymerase chain reactions (PCRs).(3)Data collection: We collected demographic and clinical data related to lung cancer, including age, sex, smoking status, tumor size, cancer staging, initial metastatic sites, neutrophil-to-lymphocyte ratio (NLR) at diagnosis, type of therapy, co-morbidities, and the Eastern Cooperative Oncology Group Performance Status (ECOG PS) score.(4)Statistical analysis: The primary endpoint was progression-free survival (PFS), while the secondary endpoint was overall survival (OS). Kaplan–Meier analysis with the log-rank test was used to analyze progression-free survival and overall survival, further stratified by different regimens of first-line treatment, including first-generation EGFR-TKI and platinum-based chemotherapy. Univariate and multivariable Cox regression analyses were used to identify the predictive factors of progression-free survival (PFS) and overall survival (OS).(5)Ethics statement: The study protocol received approval from the institutional review board at each participating center: the National Yang-Ming Chiao Tung University Hospital, Yilan, 260, Taiwan (IRB No.: 2021A022), the Far-Eastern Memorial Hospital, New Taipei City, 220, Taiwan (IRB No.: 113050-E), E-DA Hospital, Kaohsiung, 824, Taiwan (IRB No.: EMRP-110-147), and the National Taiwan University Hospital, Yunlin Branch, Yunlin, 640, Taiwan (IRB No.: 201611059RINB).

## 3. Results

(1)Patient characteristics: Among patients from these four hospitals, we excluded those with squamous cell carcinoma (SCC), those receiving only best supportive care, and those enrolled in clinical trials, leaving a total of 3825 patients. After applying additional exclusion criteria, 38 patients with EGFR exon 20 insertion mutations were identified (Figure 1). This study enrolled these 38 patients with metastatic NSCLC with *EGFR* exon 20 insertions. Their demographic data are presented in Table 1. Their median age was 65.5 years (range = 39–84 years). Most participants were female (60.5%), had never smoked (72.0%), and had an ECOG-PS score of 0–1 (81.0%). All participants had stage IV lung cancer at diagnosis, and the most common sites of distant metastases were the brain (*n* = 13, 34.2%) and bones (*n* = 13, 34.2%). A total of 4 (10.5%) patients had distant metastasis with ≥3 sites.

(2)Treatment outcomes: Regarding the initial first-line treatment, most participants were treated with *EGFR*-TKIs: 19 (50.0%) with afatinib, 8 (21.1%) with first-generation *EGFR*-TKIs, and 2 (5.3%) with osimertinib. The other nine (23.7%) patients received platinum-based doublet chemotherapy (Table 2). The rates of objective response and disease control were 31.6% and 57.9% for afatinib, 22.2% and 44.4% for platinum-based chemotherapy, and 12.5% and 37.5% for first-generation *EGFR*-TKIs, respectively. The median PFS of all was 5.15 months (0.13–40.4), the median OS was 13.0 months (0.13–76.5) (Table 3), and the survival probability curves are shown in Figure 2 and Figure 3, respectively. PFS was examined with respect to the drugs used as first-line treatments (Figure 4). The median PFS was 5.4 months (3.0–24.9) for afatinib, 5.7 months (2.1–NA) for chemotherapy, and 4.3 months (2.2–NA) for first-generation TKIs.

(3)Univariate and multivariable Cox regression analyses for predictors of PFS and OS: Predictors for PFS and OS were examined (Table 4 and Table 5). No variables significantly influenced PFS and OS.

## 4. Discussion

This study represents a comprehensive overview of the treatment and prognosis for Taiwanese NSCLC patients with *EGFR* exon 20 insertions over the past five years, effectively serving as a microcosm. It enrolled 38 patients with *EGFR* exon 20 insertions to examine factors influencing OS and PFS, including age, sex, ECOG-PS, smoking status, metastases, and brain metastasis. None of the variables significantly influenced OS.

The combination of amivantamab plus chemotherapy for locally advanced or metastatic NSCLC with documented *EGFR* exon 20 insertion mutations in the setting of first-line treatment was associated with a median PFS of 11.4 months in a previous study [9]. In our study, commonly used treatment regimens in real-world practice achieved a median PFS of 5.15 months. In Taiwan, the National Health Insurance did not reimburse the use of amivantamab until the end of 2023. Most patients with *EGFR* exon 20 insertions were given first- or second-generation TKIs as an initial treatment, which did not follow the guideline with amivantamab as first- or second-line treatment [9,10]. None of the patients in our study received amivantamab, which may have caused the difference in PFS. In a real-world registry study involving 181 patients with advanced NSCLC harboring *EGFR* exon 20 insertions, the most common first-line therapy was platinum-based chemotherapy (*n* = 111, 61.3%), which resulted in a median PFS of 6.6 months. The second most common first-line therapy was *EGFR*-TKIs (*n* = 39, 21.5%), which demonstrated a median PFS of 2.9 months for patients with *EGFR* exon 20 insertions. The duration was significantly shorter than that observed for common EGFR mutations, which had a median PFS of 10.5 months [11].

In 2017, a previous study in Taiwan [12] enrolled six patients with *EGFR* exon 20 insertions at the National Taiwan University Hospital, of which three received an *EGFR*-TKIs. Of those three patients, two showed a partial response, and one had progressive disease. Another study showed, under EGFR-TKI treatment, a median OS of 14 months (95% CI: 6-21), with a higher disease control rate being observed in complex mutations (6 of 7, 86%) compared to single mutations (16 of 40, 40%) (*p* = 0.03) [13]. A case report also documented a response to erlotinib monotherapy in a patient with this specific solitary mutation, highlighting the heterogeneity in clinical response to EGFR-TKIs among exon 20 insertion mutations [14]. Another study analyzed 1086 patients who underwent EGFR genotyping, identifying exon 20 insertions in 27 cases (2.5%). Compared to wild-type cancers, these mutations were more prevalent in never-smokers and Asian patients. The median survival for exon 20 insertion patients was 16 months, comparable to wild-type cancers but shorter than in cancers with common EGFR mutations [15].

It is crucial to recognize that *EGFR* exon 20 insertions should not be viewed as a single clinical entity, especially where insertions occurred at the A763, M766, N771, and V769 residues that confer sensitivity to afatinib [16].

Treating NSCLC patients with *EGFR* exon 20 insertions remains challenging. One major issue is detecting these mutations. The detection rate is low because of the diversity and complex structures of the insertions. According to data from Foundation Medicine, polymerase chain reactions missed 51.4% of mutations identified by next-generation sequencing (NGS). While NGS offers more comprehensive detection of *EGFR* exon 20 insertions, standardized detection methods and specifications have not yet been established. Exon 20 insertions account for 4–9% of all *EGFR* mutations [2,15,17]. A study in Taiwan examined 5608 patients with NSCLC between 2010 and 2019. Among those found to have *EGFR* mutations (*n* = 3155), 87 (2.8%) had exon 20 insertions [18]. Based on preclinical findings, the CHRYSALIS trial, a Phase I study, evaluated previously treated NSCLC patients with EGFR exon 20 insertion mutations. The results showed a median overall survival of 22.8 months, median progression-free survival of 8.3 months, and an objective response rate of 40% [19]. The Phase I CHRYSALIS study showed promising results, leading to the Phase III PAPILLON trial, which compares amivantamab plus chemotherapy to chemotherapy alone as a first-line treatment for NSCLC patients with EGFR exon 20 insertion mutations [9]. The trial enrolled 308 patients, comparing carboplatin plus pemetrexed with or without amivantamab. The combination showed higher ORR (73% vs. 43%), longer DoR (9.7 vs. 4.4 months), and extended PFS (11.4 vs. 6.7 months). At 18 months, PFS was 31% with amivantamab vs. 3% with chemotherapy alone. Median OS favored the combination, with 24.4 months for chemotherapy, and the time not yet reached for amivantamab. Integrating exon 20 insertion-targeted therapies into first-line treatment is reshaping NSCLC management. Trials like the PAPILLON trial have assessed their early use, emphasizing the need for effective combinations to enhance outcomes. Amivantamab with chemotherapy offers a comprehensive approach to tackling this aggressive mutation, highlighting the demand for ongoing research and robust clinical trials. The Phase III EXCLAIM-2 trial compared first-line mobocertinib to platinum-based chemotherapy in NSCLC patients with EGFR exon 20 insertions. Results showed similar efficacy, but mobocertinib failed to surpass chemotherapy in PFS, leading to its withdrawal as a treatment option [20]. The WU-KONG 1 and 2 trials are assessing sunvozertinib in metastatic NSCLC with EGFR or HER2 mutations. A pooled analysis showed a 50% ORR in 56 patients with EGFR exon 20 insertions across dose levels [21]. The WU-KONG 6 study reported a 60.8% ORR for sunvozertinib in Chinese NSCLC patients with EGFR exon 20 insertions [22].

While its sample size was modest, our study combined data from the latest databases of four large hospitals in Taiwan, enrolling more patients than previous studies. Due to National Health Insurance, most patients with *EGFR* exon 20 insertions in Taiwan may receive TKIs as their first-line treatment. Our experience has shown that TKIs could be considered as a therapy. Whether afatinib plays a role in this context for some patients depends on the specific nature of the mutation, and remains to be determined.

Regarding factors influencing PFS and OS, our study found no significant difference among smokers and never-smokers (Table 4 and Table 5). However, our findings differ from previous studies [23,24,25], which suggested that those with less cigarette exposure had better outcomes. However, our findings differ from previous studies [23,24,25], which suggested that those with less cigarette exposure had better outcomes. In our study, cigarette exposure time was not well documented, nor was lung function, which may have biased our results. The observation of decreased PFS when TKIs were used in first-line therapy remains consistent with previous studies [26,27].

## 5. Limitation

This study has several limitations that should be acknowledged. First, the cohort study design may lack sufficient clarity in certain aspects, potentially impacting the ability to draw fully unbiased statistical conclusions. Specifically, limitations in the selection criteria and data collection process could introduce bias, which may affect the generalizability of the findings.

While efforts were made to minimize these biases through rigorous data analysis and adjustments, we recognize that a more refined study design could further strengthen the validity of the results. Future research should consider revising the study framework and collecting additional data to validate and extend these findings. By addressing these limitations, subsequent studies could offer more robust evidence, enhancing the reliability and applicability of these conclusions in broader contexts.

While this study enrolled more patients than previous studies, the sample size of 38 remains relatively small. This limitation suggests that the findings may not fully represent the broader population of patients. Consequently, further studies with larger sample sizes are essential to validate these results and enhance the reliability of the conclusions drawn. Larger cohorts would provide a more comprehensive understanding of the effects and implications of the treatment, ultimately contributing to improved patient care and outcomes in this area of research. Continued exploration is necessary to confirm these preliminary findings and strengthen the evidence base. *EGFR* exon 20 insertions are rare, and some patients do not undergo NGS for financial reasons, making these mutations more challenging to identify.

Furthermore, detection accuracy varies across detection methods, and the four hospitals participating in our study do not use standardized methods and tools for detection. This limitation also applied to other studies, and the use of various local testing methods may have introduced some undetected biases. Among the more than 200 cases with *EGFR* exon 20 insertions, only 42 underwent NGS testing [16]. Due to the small sample size, our study could only be observational; thus, a specific treatment design or sequence could not be expected.

## 6. Conclusions

This study provides an in-depth analysis of the treatment outcomes and prognostic trends in patients with *EGFR* exon 20 insertions in Taiwan over the last five years. Among those with *EGFR* exon 20 insertions, median PFS was 5.15 months, and OS was 13 months, with 76.3% of cases treated with EGFR-TKIs as initial therapy. Notably, former smokers exhibited significantly lower OS than current and never smokers, whereas there were no discernible differences in PFS among these groups. The potential role of afatinib in this context for specific patients is contingent on the exact nature of their mutation and has yet to be established.

## Figures and Tables

**Figure 1 curroncol-32-00174-f001:**
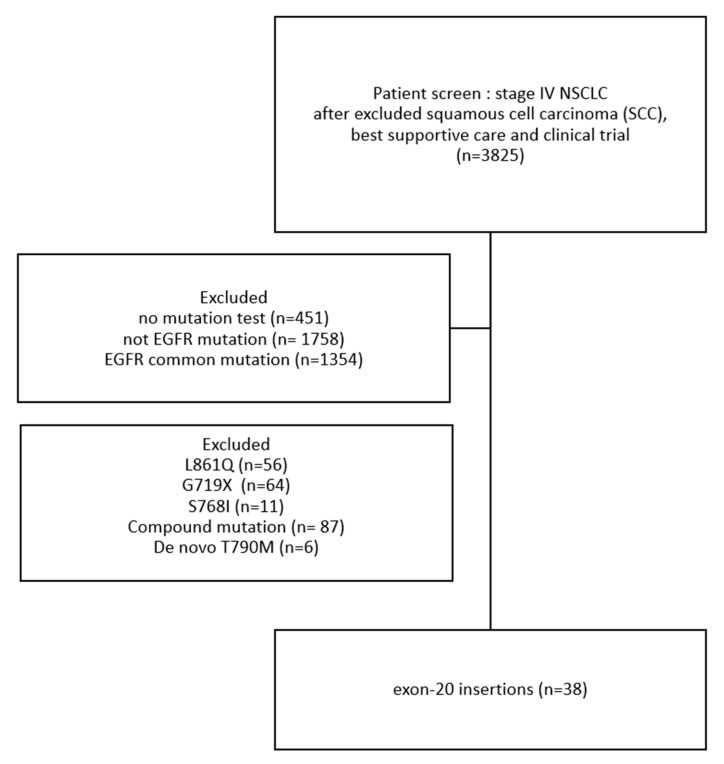
Flowchart of Patient Selection and Mutation Screening in Stage IV NSCLC.

**Figure 2 curroncol-32-00174-f002:**
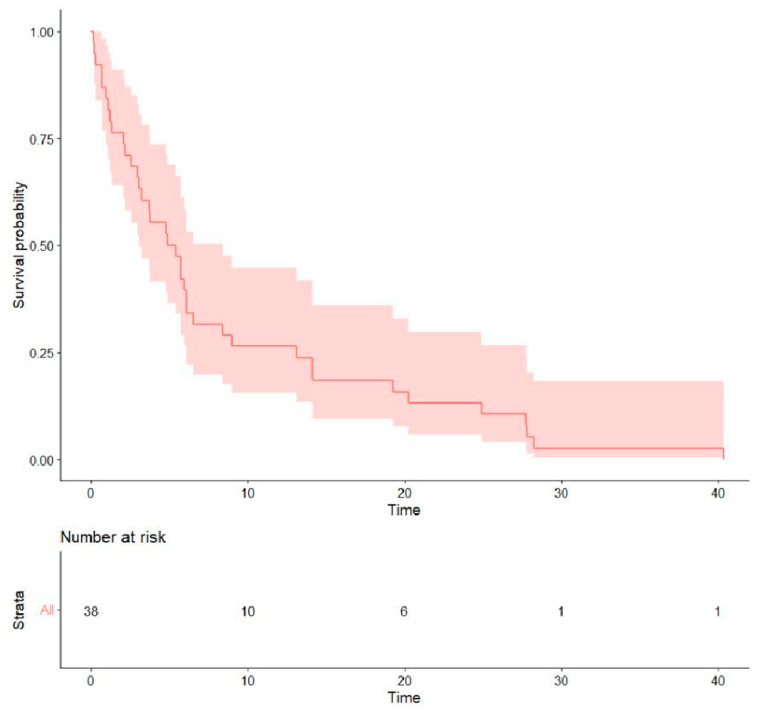
Progression-free survival following initial first-line treatment.

**Figure 3 curroncol-32-00174-f003:**
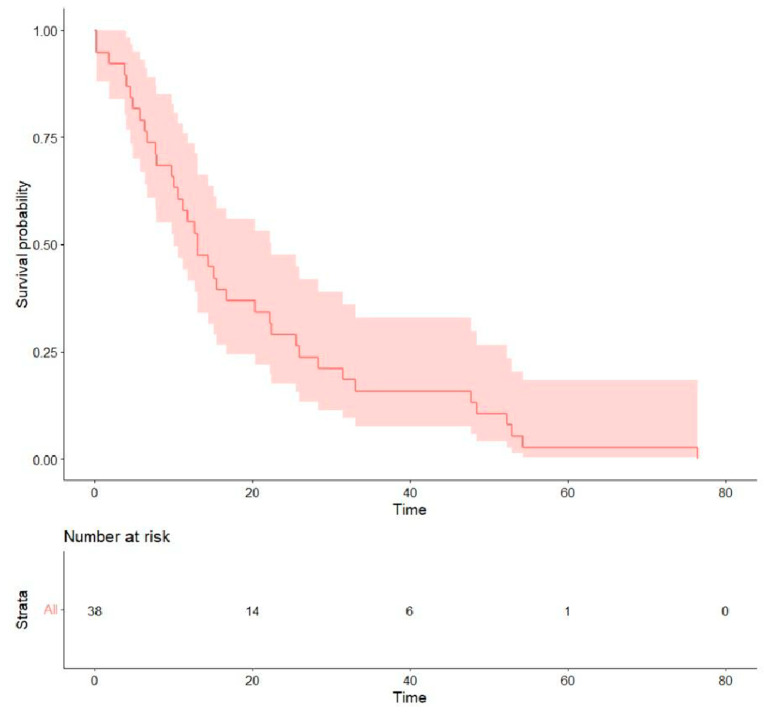
Overall survival following the initial first-line treatment.

**Figure 4 curroncol-32-00174-f004:**
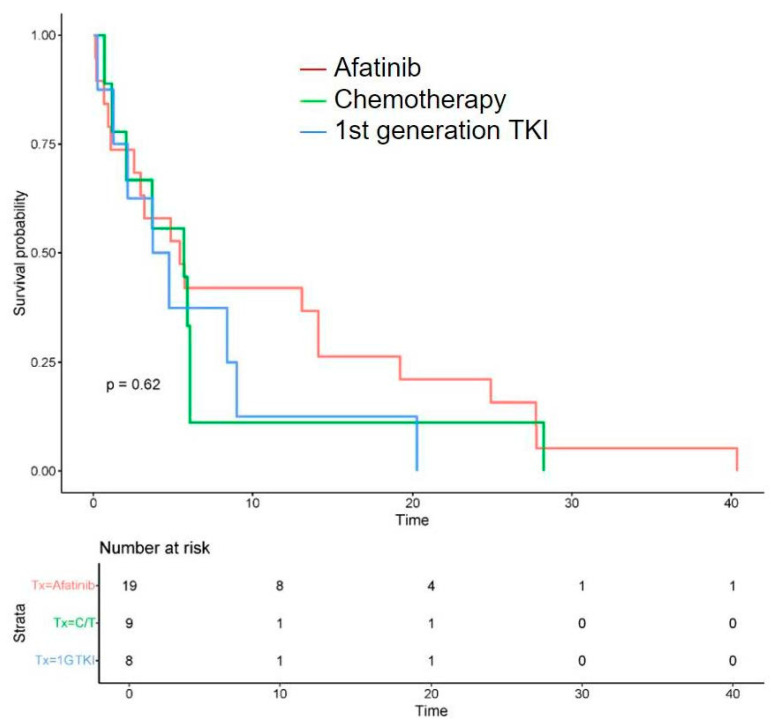
Progression-free survival (PFS) with different systemic treatment approaches against advanced NSCLC harboring *EGFR* exon 20 insertions. PFS was 5.4 months (3.0–24.9) for afatinib, 5.7 months (2.1–NA) for chemotherapy (C/T), and 4.3 months (2.2–NA) for first-generation (1G) TKIs.

**Table 1 curroncol-32-00174-t001:** Clinical Characteristics of enrolled patients with NSCLC harboring *EGFR* exon 20 insertions.

Number of Patients	38		
	Median (Range)		
Age (years),	66 (39–84)		
BSA	1.61 (1.32–2.12)		
Target tumor size	3.4 (0.6–11.4)		
CEA level	35.92 (0.45–1037.26)		
		*n*	%
Age			
	<65	17	44.7
	>65	21	55.3
Gender			
	Female	23	60.5
	Male	15	39.5
ECOG PS			
	0	18	47.4
	1	13	34.2
	2	4	10.5
	3	3	7.9
Smoking status			
Smoking			
	Yes	14	36.8
	No	24	63.2
Metastasis			
Site	≥3	4	10.5
	<3	34	89.5
Brain	YES	13	34.2
	NO	25	65.8
Lung	YES	7	18.4
	NO	31	81.6
Malignant effusion	YES	11	28.9
	NO	27	71.1
Liver	YES	0	0.0
	NO	38	100.0
Bone	YES	13	34.2
	NO	25	65.8
Adrenal gland	YES	1	2.6
	NO	37	97.4
Co-morbidity			
	YES	16	42.1
	NO	22	57.9
Hypertension			
	YES	11	28.9
	NO	27	71.1
Diabetes mellitus			
	YES	2	5.3
	NO	36	94.7
Liver disease			
	YES	1	2.6
	NO	37	97.4
COPD			
	YES	4	10.5
	NO	34	89.5
Neutrophil/lymphocyte ratio			
(Blood)	≥5	14	36.8
	<5	24	63.2

NSCLC: non-small-cell lung cancer; CEA: carcinoembryonic antigen; ECOG PS: Eastern Cooperative Oncology Group Performance Status; COPD: chronic obstructive pulmonary disease. BSA: $\sqrt{Body\;weight\;(kg)\times Body\;high\;(cm)/3600}.$

**Table 2 curroncol-32-00174-t002:** Distribution of first-line treatment modalities.

	*n*	%
Afatinib	19	50.0
Osimertinib	2	5.3
Chemotherapy	9	23.7
First generation TKIs	8	21.1

TKI: tyrosine kinase inhibitor.

**Table 3 curroncol-32-00174-t003:** Progression-free survival (PFS) and overall survival (OS) in those who received treatment.

	Median (Range)
PFS (months)	
	5.15 (0.13–40.4)
OS (months)	
	13.0 (0.13–76.5)

PFS: progression-free survival; OS: overall survival.

**Table 4 curroncol-32-00174-t004:** Univariate and multivariable Cox regression analyses for predictors of PFS.

	Univariate		Multivariable	
Variable	HR (95% CI)	*p* Value	Adj. HR (95% CI)	*p* Value
Age				
<65	Ref.		Ref.	
≥65	0.98 [0.51–1.90]	0.963	1.12 [0.46–2.72]	0.801
Gender				
Female	Ref.		Ref.	
Male	1.35 [0.69–2.66]	0.381	1.58 [0.68–3.69]	0.290
ECOG PS				
0–1	Ref.		Ref.	
≥2	1.45 [0.63–3.34]	0.405	2.16 [0.70–6.66]	0.179
Smoking status				
No	Ref.		Ref.	
Yes	1.17 [0.85–1.61]	0.360	1.15 [0.81–1.62]	0.433
Metastatic sites				
<3	Ref.		Ref.	
≥3	1.35 [0.46–3.97]	0.580	2.39 [0.61–9.42]	0.212
Brain metastasis				
No	Ref.		Ref.	
Yes	0.96 [0.49–1.92]	0.918	0.73 [0.31–1.72]	0.468
NLR				
No	Ref.		Ref.	
Yes	1.06 [0.54–2.09]	0.869	0.83 [0.38–1.82]	0.638

PFS: progression-free survival; Adj. HR: adjusted hazard ratio from the multivariable analysis; CI: confidence interval; ECOG PS: Eastern Cooperative Oncology Group Performance Status; NLR: neutrophil/lymphocyte ratio.

**Table 5 curroncol-32-00174-t005:** Univariate and multivariable Cox regression analyses for predictors of OS.

	Univariate		Multivariable	
Variable	HR (95% CI)	*p* Value	Adj. HR (95% CI)	*p* Value
Age				
<65	Ref.		Ref.	
≥65	1.15 [0.60–2.21]	0.667	1.48 [0.53–4.09]	0.452
Gender				
Female	Ref.		Ref.	
Male	1.26 [0.65–2.43]	0.500	1.30 [0.46–3.69]	0.617
ECOG PS				
0–1	Ref.		Ref.	
≥2	1.67 [0.71,3.90]	0.260	1.83 [0.60–5.60]	0.289
Smoking status				
No	Ref.		Ref.	
Yes	1.10 [0.57–2.15]	0.772	1.73 [0.63–7.72]	0.286
Metastatic sites				
<3	Ref.		Ref.	
≥3	2.11 [0.72–6.15]	0.173	4.24 [0.95–19.03]	0.059
Re-biopsy				
No	Ref.		Ref.	
Yes	0.45 [0.18–1.13]	0.089	0.57 [0.16–1.40]	0.176
Brain metastasis				
No	Ref.		Ref.	
Yes	1.14 [0.57–2.26]	0.712	0.71 [0.30–1.70]	0.445
NLR				
No	Ref.		Ref.	
Yes	1.23 [0.63–2.39]	0.548	1.09 [0.50–2.36]	0.838

OS: overall survival; Adj. HR: adjusted hazard ratio from the multivariable analysis; CI: confidence interval; ECOG PS: Eastern Cooperative Oncology Group Performance Status; NLR: neutrophil/lymphocyte ratio. Re-biopsy: biopsy performed after the failure of first-line or subsequent-line treatments.

## Data Availability

The data that support the findings of this study are available from the corresponding author upon reasonable request.

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
