# Peer review of "Unraveling Survival Determinants in Patients with Advanced Non-Small-Cell Lung Cancer with EGFR Exon 20 Insertions"

_curroncol, 2025, doi:10.3390/curroncol32030174_

Round 1
Reviewer 1 Report
Comments and Suggestions for Authors
Wang et al. investigate patients with advanced non-small cell lung cancer (NSCLC) harboring EGFR exon 20 insertions, utilizing data from multiple centers to analyze the effects and clinical significance of this rare EGFR mutation on responses to EGFR-TKIs. However, there are notable issues with the data analysis, making the conclusions drawn from this study less reliable.
Major points:
- The conclusion that EGFR-TKIs could be considered a viable treatment option for patients with EGFR exon 20 insertions is questionable. The reported efficacy does not appear to be sufficiently strong to justify this recommendation. Please provide additional justification or reconsider the strength of this conclusion.
- In the Introduction, “According to data from the Ministry of Health and Welfare, lung cancer was the leading cause of cancer-related deaths in Taiwan in 2019.” As it is now 2025, this data should be updated to the most recent available statistics.
- Why did the study exclusively enroll patients with stage IV NSCLC? Were patients with earlier-stage disease excluded for a specific reason?
- In Tables 4 and 5, what is meant by “adjusted HR” in the multivariable analysis? There appears to be missing data in this column. If the analysis was not performed, please clarify why. Otherwise, ensure that the table is complete and accurately reflects the statistical findings.
- The term “AIC” is mentioned in the table, but there is no description of it in the Methods section. Please define AIC and explain its role in your analysis. If it was used for model selection, provide details on how it was applied. Additionally, based on the structure of your multivariable analysis, there appears to be some misunderstanding in its application. Otherwise, the multivariable analysis column would not be empty.
- How do you define “current smoker” and “former smoker” in your study? Instead of these categories, why not simplify the classification into “smoker” and “non-smoker”? Alternatively, a categorization based on the duration or intensity of smoking (e.g., pack-years) could provide more meaningful insights.
Author Response
-
Revised Letter
Thank you for all your suggestion. We had made a correct manuscript according to the reviewer’s suggestion. All black word version is final manuscript and red word version preserved the change. Thank you very much for reviewing our text and gave our change for publication. Here as follows is the detail response according to the reviewer’s suggestion.

Reviewer 2 Report
Comments and Suggestions for Authors
Authors have performed analysis on treatment outcomes and prognosis in NSCLC patients with EGFR Exon 20 mutation. Study is well done and informative.
It will be nice to have a larger follow-up study with a larger cohort, but I understand it can be challenging, since this is a rare mutation.
So far study looks good.
It can be published.
Author Response
Thank you for your thoughtful feedback and recognition of our study. We appreciate your understanding of the challenges in conducting larger follow-up studies due to the rarity of this mutation. Your positive remarks and support for publication are truly encouraging.
Reviewer 3 Report
Comments and Suggestions for Authors
Abstract: summarizes adequately the article
Introduction: good background contextualization
Patients and methods
Line 65: what do you mean by medical center? Central hospital, cancer center…?
Elucidate the inclusion criteria and if there were any exclusion criteria. For example, all consecutive patients with stage IV NSCLC, who had EGFR exon 20 insertion mutations, diagnosed between January 2016 and December 2020 were included? Patients referred only to best supportive care and patients treated in clinical trial, were they included?
Include information about the method used to determine the mutational status?
Results
Consider adding a diagram of patients, exposing how many EGFR patients were identified, from those, how many with common and uncommon mutations, and in the end the 38 patients with exon20 ins. That way the reader has a real-world insight of the incidence of EGFR Exon 20 Insertions in Taiwan.
Table 1
Consider changing weight and height by body surface.
Consider including only relevant comorbidities, for example, if none of the patients had chronic kidney disease, why mention it?
Table 3
Consider omitting this table. The data is clearly exposed in the text. If you agree with omitting this table, review the sentence line 105 – 106 “The participants’ median OS and PFS are shown in Table 3, and the survival probability curves are shown in Figures 1 and 2, respectively”.
Line 113-114: “Table 3. Progression-free survival (PFS) and overall survival (OS) in those who received treatment.” Why not in all population included, since all received treatment?
Table 5: Contextualize what do you mean by Re-biopsy.
Discussion
Line 154-155 “… which did not follow the guideline with amivantamab as first- or second-line treatment 9.” Review the reference. Ref. 9 is clinical trial Papillon, consider changing it for a guideline, such as Asian guidelines, ESMO, NCCN…
Line 167: “That study also reviewed the published data (2-4,12-14) concluding that patients with 167 EGFR exon 20 insertions treated with an EGFR-TKIs (n = 11) had a response rate of 73%.” Who are the n=11? Are the patients identified in those studies with exon 20 ins? It is not clear in the text. Consider complementing information with a table or clarifying it in the text.
Line 192-193: “… when TKIs were used beyond first-line therapy…” don’t understand this. The PFS you showed was for first line. Please clarify.
Limitations: the two main limitations are mentioned, the sample size and method used for detection of EGFR mutations
Conclusion: adequate for the findings.
Author Response
Thank you for all your suggestion. We had made a correct manuscript according to the reviewer’s suggestion. All black word version is final manuscript and red word version preserved the change. Thank you very much for reviewing our text and gave our change for publication. Here as follows is the detail response according to the reviewer’s suggestion.

Round 2
Reviewer 1 Report
Comments and Suggestions for Authors
The authors have addressed the issues raised in the previous review. I have no further concerns.